# Impact of Physician Practice Racial Composition on Patient Demographics

**DOI:** 10.3390/healthcare13182255

**Published:** 2025-09-09

**Authors:** Gnankang Sarah Napoé, Hyagriv N. Simhan, Lara S. Lemon

**Affiliations:** 1Department of Obstetrics, Gynecology and Reproductive Sciences, University of Pittsburgh School of Medicine, Pittsburgh, PA 15213, USAlemonl@upmc.edu (L.S.L.); 2UPMC Magee-Womens Hospital, Pittsburgh, PA 15213, USA; 3Magee-Womens Research Institute, Pittsburgh, PA 15213, USA

**Keywords:** Black physicians, physician–patient relations, health services accessibility, minority health, gynecology

## Abstract

**Background:** Designing practices to better serve Black patients is necessary to decrease health disparities in America. Objective: To understand the impact of physician practice racial composition on patient demographics in a gynecology practice in the United States. We hypothesized that there will be an increased proportion of Black patients seen by all physicians within a practice by adding Black physicians to that practice. Design: This is a retrospective study comparing patient demographics of three subspecialty gynecology practices: Practice A, with two Black physicians added to the staff during the study period, and Practice B and C, without any Black physicians. **Methods:** We compared patient demographics by practice over time, including race (Black and White), insurance status (public vs private), and area deprivation index (ADI) as a proxy for socioeconomic status. **Results:** During the study period, there was a statistically significant increase in Black patients in practice A (slope = 0.0029; *p* < 0.001), while the proportion trend of Black patients decreased or remained flat in Practice B (slope = −0.0020; *p* = 0.027) and C (slope = −0.0010; *p* = 0.23), respectively. While Black physicians saw more Black patients than nonBlack physicians in Practice A, the proportion of Black patients seen by nonBlack physicians, though modest, steadily increased during the study period. Practice A saw patients with higher ADI and more patients with public insurance. Black physicians in practice A saw more publicly insured patients than nonBlack physicians. There was no difference in patients’ ADI whether they were seen by Black or nonBlack physicians in Practice A. **Conclusions:** The addition of Black physicians to a practice is associated with increasing the proportion of Black patients seen by both Black and nonBlack physicians in that practice.

## 1. Introduction

Health disparities in maternal mortality and morbidity, neonatal morbidity, contraceptive use, vaccination rates, reproductive cancer outcomes, and other gynecologic conditions have been documented in the United States (US) [1]. Access to care has been proposed as a potential contributor to those disparities. Physician patient racial concordance has been linked to seeking timely care and better healthcare utilization, particularly for Black and White patients [2,3]. In addition, Black patients are more satisfied with their care when they have Black physicians [4]. More important than patient satisfaction, health outcomes are improved for some Black patients when they have a racially concordant physician [5]. There are not enough underrepresented physicians for racially concordant care to be a viable strategy in the short term, and any particular patient is likely to need care from multiple physicians, including subspecialists. Strategies to improve the diversity of the physician workforce in the US have been proposed [6,7]. Specific steps, including prioritizing diversity, improving the learning and work environment for underrepresented physicians, considering the diversity of the applicant pool in relation to the population served, and mentoring, have been proposed [6,7]. However, data suggest that even with doubling medical school matriculants from underrepresented backgrounds, it will take 60–90 years to achieve adequate representation of physicians from underrepresented racial and ethnic groups [8]. Therefore, it is important to understand how to improve access to care for Black patients that does not involve Black patients always having to see a Black physician—for the good of Black patients and the population at large. The indirect cost of health care disparities is borne by all Americans [9]. Chronic illness, premature deaths, and loss of productivity in Black patients account for the majority of the costs in the US [9].

A population-level study indicated that Black patients who lived in counties with Black physicians had higher life expectancies than those living in counties without Black physicians—even if they were not receiving care from a Black physician [10]. Indeed, it is improbable to improve population-wide health outcomes if health outcomes in underserved populations are ignored. This leads us to wonder if the addition of Black physicians to a practice would change the demographics of the entire practice. We hypothesized that the addition of Black physicians to a practice would lead to an increase in the number of Black patients seen in that practice through trust-building. Because prior studies have reported that Black primary care physicians see more socioeconomically disadvantaged patients, our secondary aim was to determine whether the addition of Black physicians to this subspecialty gynecology practice would lead to seeing more socioeconomically disadvantaged patients. Socioeconomic disadvantage was assessed by proxy using area deprivation index (ADI) and insurance status.

## 2. Methods

This is a retrospective study of all persons with outpatient office visits (inclusive of face-to-face and telemedicine) with physicians at three subspecialty gynecology practices within a tertiary care hospital, from 1 January 2015, through 31 December 2024. We elected to use this 10-year period for the study because it included two years of visits prior to the addition of any physicians of color, then the addition of one Black doctor in 2017, followed by the addition of a second Black doctor in 2019 in Practice A. Practice B and C were predominantly White; however, in August 2024, Practice C added a Black doctor. Therefore, we have only included Practice C’s patient data through July 2024.

In our primary analysis, we compared the three practices to determine if changes in the demographic composition of the patient population were due to the addition of Black physicians rather than population differences within the region, as these practices are co-located in one hospital and therefore serve the same region. We chose two gynecology practices that provide quality-of-life services to patients, including management of pelvic floor disorders and menopause care (Practices A and B). To minimize the selection bias that may occur with only comparing two practices, we added a third practice focused on reproductive infertility (Practice C). The choice of practices was to decrease any potential confounders that may be perceived with essential conditions such as obstetric or oncologic care, where patients must seek clinical care regardless of their preferences. Physician race was determined from a publicly available website at the University of Pittsburgh School of Medicine with a list of physicians from racial backgrounds considered underrepresented in medicine (https://www.medschool.pitt.edu/, accessed on 5 November 2024). We compared racial demographics data for patients seen in all three practices.

We accessed data through the health system’s Enterprise Data Warehouse (EDW). The EDW stores all discrete documentation from the electronic health record, including but not limited to patient demographics, clinical characteristics, and diagnoses.

A patient’s address at the time of every visit within our system is mapped to the exact location (i.e., zip+4 code), and the corresponding ADI for this neighborhood is included in the EDW. ADI is an accepted marker of neighborhood-level advantage. This index ranks all neighborhoods within the United States from 1 to 100, with 1 representing the most advantaged neighborhoods and 100 indicating neighborhoods with the most disadvantage. While ADI does not reflect individual patient circumstances, its measure of neighborhood-level advantage impacts patients’ health outcomes [11].

This study was approved by the University of Pittsburgh’s Institutional Review Board.

### Analysis

The data was structured such that a patient contributes once to the denominator of each year in which they had a visit with an attending physician in any practice. For example, a White patient with 3 visits in 2015, 1 visit in 2016, and none in 2017 will be represented once in 2015 and 2016 but not counted in 2017. If a patient was seen in each practice, they will be represented in each.

Using self-reported race documented at the time of first visit each year, we graphed the proportion of the patient population served by the practice that self-reported Black or White race over time. We limited the comparison to the Black race and the White race, as all other races combined comprised less than 5% of the population. In addition, the group was heterogenous, including more than 10 distinct self-reported races, and therefore these numbers were too low to compare. Using trend lines to determine the slopes and regressions to calculate the *p*-values, we demonstrated if the change in racial distribution in each group was statistically significant over time, defining *p*-values less than 0.05 as statistically significant. A higher gradient indicates a steeper slope, with positive numbers indicating an increase and negative numbers reflecting a decrease in the proportion.

Next, we modeled the likelihood of a patient being Black within each practice, using the visit year as the exposure and 2015 as the baseline. These logistic regression models were adjusted for the patient’s age, insurance type, and ADI along with the total number of visits that patients had outside of Practice A, B, and C that year. A person could contribute to each year once, regardless of the number of visits within the clinics. For information that changed over time: we used the highest age and ADI annually, and if the patient ever had public insurance that year, they were categorized in public insurance. As a marker of healthcare utilization, we totaled all outpatient visits within and outside of practices of interest for each patient during the study year.

We then explored patterns of patient demographics within Practice A by physician race. The proportion of Black patients was compared from 2019 to 2024 when there were at least 2 Black physicians to avoid identifiable patient demographics for any one physician. Grouping both Black physicians compared with all others, we graphed the proportion of the annual patient population by race. This analysis was to explore if the entire group began to see more Black patients. We again include *p*-values for the change in racial demographics over time to indicate statistical significance.

In sensitivity analyses limited to practices A and B, we compared other markers of social determinants of health, including ADI and insurance type (public vs. commercial), and how they changed over the study period. To explore if these factors changed independent of race, we repeated these trends excluding Black patients. We visualized the proportion of deliveries at our hospital that self-report Black versus White race annually since 2015. This was done to reduce selection bias by exploring the demographics of a population not seeking “elective” care in our region.

Finally, we calculated the median number of office visits outside of Practice A and B over time. Outside office visits were included for the year that the visit with Practice A or B occurred. We demonstrated these utilization rates by patient race, then provider race within Practice A, and by patient race within Practice B.

## 3. Results

Our population of Black or White patients seen in Practice A was comprised of 16,283 individuals, generating 33,923 visits in Practice A throughout the study period. In Practice B there were 10,263 individuals generating 25,478 visits; in Practice C 9497 individuals generated 25,282 visits. The number of patients seen annually per practice was included in Table 1. ADI, missing for <5% of visits, was imputed with the median ADI for the practice. This was the only variable with missing data, and results did not vary when included with missingness compared with imputed values.

The proportion of patients with self-reported Black race markedly increased in Practice A over time (slope = 0.0029; *p* < 0.001); this upward trend was not apparent in Practice B (slope = −0.0020; *p* = 0.027) nor C (slope = −0.0010; *p* = 0.23) (Figure 1). Regressions show a similar trend, with the increased likelihood of seeing a Black patient linearly increasing over time in Practice A as a whole. A visit in 2024 was 49% more likely to be with a Black patient compared with 2015 (adj OR: 1.49; 95% CI: 1.17, 1.88). The proportion of Black patients began to statistically significantly increase in 2021, two years after the addition of the second Black physician. There was no significant increase in Black patients in Practice B or C during the study period. In fact, in practice B, there was a statistically significant decrease in the likelihood of seeing a Black patient in 2024 compared with 2015 in Practice B [adj OR 0.68 (95%CI: 0.51, 0.91)].

We then evaluated the proportion of Black patients seen by Black and nonBlack physicians, independently, within Practice A from 2019 to 2024. Figure 2 demonstrates that the proportion of Black patients increased for both groups of physicians, though statistical significance was limited to Black physicians (slope = 0.0074; *p* = 0.035) compared with nonBlack physicians (slope = 0.0021; *p*-value = 0.26) (Figure 2).

Sensitivity analyses demonstrated that the number of deliveries with self-reported Black race decreased significantly over the study period (slope = −0.0062; *p*-value < 0.001). In 2015, 23% of all deliveries self-reported Black race compared with 19% in 2024.

Finally, trends in median ADI and insurance type for each patient population over time remained flat for both Practice A and Practice B. Practice A compared with Practice B had notably more visits with patients at a higher ADI or more socioeconomic disadvantage (64 vs. 48) and with public insurance (40% vs. 16.3%), respectively, during the study period. Those flat trends were consistent when removing Black patients from the population analyzed. There was a higher proportion of publicly insured patients seen by Black physicians than nonBlack physicians in practice A, but those trends slightly increased over time for both groups.

In our final sensitivity analyses exploring outpatient utilization beyond the practices of interest, we found statistically significant increases in median office visit count annually for those patients being seen in Practice A or B. Results were consistent regardless of patient race or when stratified by provider race with Practice A. Specifically, for patients seen in Practice A, the median number of visits within our system increased from 4 to 8 and 3 to 5 for Black (*p* = 0.002) and White patients (*p* = 0.002), respectively, from 2015 to 2024.

## 4. Discussion

In this study of three subspecialty gynecology practices, the addition of Black physicians to a practice was associated with a statistically significant increase in the proportion of Black patients over time in that practice. The increase in proportion of Black patients seen in the practice was also notable for the nonBlack physicians in the practice, though this was not statistically significant. This occurring while there was a flat trend in the proportion of Black patients in the practices without Black physicians, along with a decrease in Black deliveries in the region, suggests that the increase in Black patients within the practice with Black physicians is unlikely to be attributable to a general change in the demographics of the geographic region. The increase in proportion of Black patients seen by all physicians suggests that practices that have more Black physicians may be more likely to serve Black patients. Prior studies have suggested that Black patients feel more comfortable when there are more people of color in a practice [12]. This may be more notable in regions where there is less trust of physicians. Our study was conducted in Pittsburgh, which was previously shown to have extreme disparities in health outcomes between Black and White women [13]. It is possible that some disparities may be related to lack of healthcare seeking if Black patients anticipate additional instances of discrimination [14].

While our study did not seek to understand how the addition of Black physicians led to the increase in Black patients, we speculate that the increase in Black patients seen by Black physicians likely leads to more patient-to-patient referrals for the practice. In addition, in academic settings, Black physicians are more likely to focus on a research agenda with community participation [15]. This can lead to improving the patient population’s impression of the practice, which might appeal to new patients. Any community outreach performed by Black physicians may also potentially lead to attracting more Black patients to the practice. It is also possible that, as seen in the business field where more diverse teams are more successful, likely due to more objectivity, practices with physicians of different racial identities may serve minoritized patients better [16]. This should be an area for future research.

We found no change in the proportion of socioeconomically disadvantaged patients with the presence of Black physicians in the practice. This may be because these are subspecialties focusing on quality-of-life care. There were differences in ADI and insurance type between the two practices even at baseline. Practice A manages pelvic floor disorders, including pelvic organ prolapse, urinary incontinence, and pelvic pain. Whereas Practice B’s focus is on midlife care, including management of vasomotor symptoms, abnormal uterine bleeding associated with perimenopause, sexual dysfunction, and urinary incontinence. The ADI differences may be due to the inherent differences of conditions treated by each of those practices.

Since prior studies have suggested an increase in healthcare utilization among patients of Black physicians, we assessed healthcare utilization in the practices [3]. In our study, patients from all practices, regardless of race, had an increase in outpatient office visits during the study period. We surmise that the potential drivers of the notable increase in outpatient office visits for all practices include the growth of the hospital system through acquisition of private outpatient offices and smaller hospital systems and the uptake of electronic medical records throughout the system. In addition, there have been hospital-wide initiatives to increase access to care and the uptake of telemedicine, increasing the availability of different specialties to patients formerly limited by geography [17]. Our study demonstrated a higher proportion of publicly insured patients seen by Black physicians. This is similar to findings in primary care [18,19].

While the field of obstetrics and gynecology in the US has recognized its disparities and acknowledged the need to increase the number of minoritized physicians in our workforce, to our knowledge, this is the first study in gynecology to assess whether practice racial makeup impacts the racial demographics within that practice [20]. Given the dearth of Black physicians in the US, practices aiming to improve the care of marginalized patients in order to improve the health and productivity of the entire community may achieve better results by adding Black physicians to their practice.

In our study, it took two years following the addition of a second Black physician to begin noticing a statistically significant increase in the proportion of Black patients. This may be due to the fact that when patients see a lone Black physician in a large practice, they may perceive their presence as a sign of tokenism [21,22].

We hoped to find a statistically significant increase in Black patients being cared for by nonBlack physicians in the diverse practice, but this increase was not significant. We may need more study years or a larger percentage of Black physicians in a practice to notice a statistically significant difference. Further research should be done in this area in more diverse communities.

Our study strengths include the robust length of the study that facilitated the ability to notice a statistically significant increase in Black patients, the use of the data warehouse, and the ability to pick three women’s health practices, comprising two quality-of-life practices, to include in the data analyzed. Nonetheless, the fact that the practices do not provide the same type of clinical care was a limitation, and our large sample size contributes to statistical significance even for small effect sizes. The higher socioeconomic status among patients from Practice B is most likely not due to the physicians in that practice but rather because the experience with menopause is influenced by class [23]. Women from lower socioeconomic classes are less likely to receive treatment for menopause symptoms [24]. However, because Black women have more bothersome vasomotor symptoms, we would have expected a higher proportion of Black patients in Practice B if Black physician presence did not influence care-seeking [25]. The fact that the proportion of Black patients did not increase during the study period for Practices B or C and the overall decrease in births by Black patients at the hospital seem to strengthen our conclusion that the increase in Black patients was associated with the presence of Black physicians in Practice A.

A limitation of our methods is that we modeled race knowing it is not a modifiable outcome. We did this to demonstrate statistical significance and association, acknowledging the inherent lack of causality. Another limitation is that while Black physicians’ race was known based on their inclusion on the university website of minority physicians and their own self-report, self-identified race was not obtained for nonBlack physicians. The three practices were in a large tertiary academic center in an urban setting, so the trends noted in those practices may not be generalizable to different practice settings. Because this study is only focused on physicians, we cannot speak to whether the same patterns would be seen with other healthcare providers. Based on the nature of the study being a database review, we were unable to assess what factors led to an increase in visits. We assume that Black participants were more trusting of the practice in general, as it had Black physicians. Previous research showed that Black patients are more likely to discuss sensitive medical conditions with health care providers when there are more minoritized people on staff [12]. This is consistent with a proposed framework on care seeking that argues that therapeutic relationships are improved when there is patient trust of healthcare providers [26].

In addition, because our study focused on the Black race, we do not know whether the same trends would be seen with other minoritized groups. Finally, we made an assumption that an increased number of visits leads to better health outcomes. Future studies need to evaluate health outcomes of practices seeing an increasing number of Black patients.

## 5. Conclusions

The addition of Black physicians to a subspecialty gynecology practice led to an increase in Black patients for this practice. Healthcare organizations seeking to care for more Black patients should hire more Black physicians.

## Figures and Tables

**Figure 1 healthcare-13-02255-f001:**
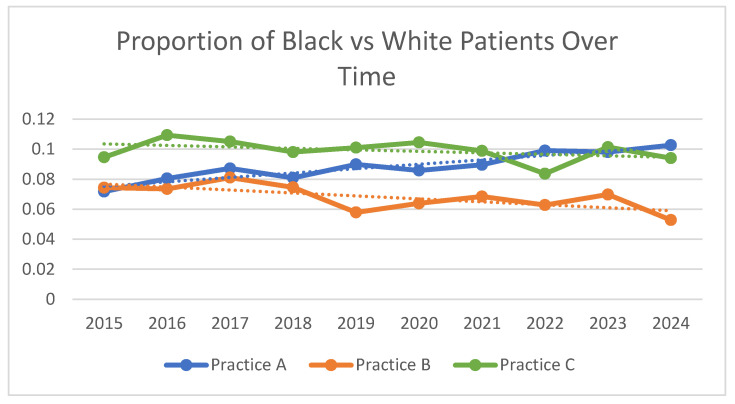
Percent of patient population self-reporting Black versus White race at first visit annually over time; comparing Practice A to Practice B. Dotted lines represent linear trend lines by Practice.

**Figure 2 healthcare-13-02255-f002:**
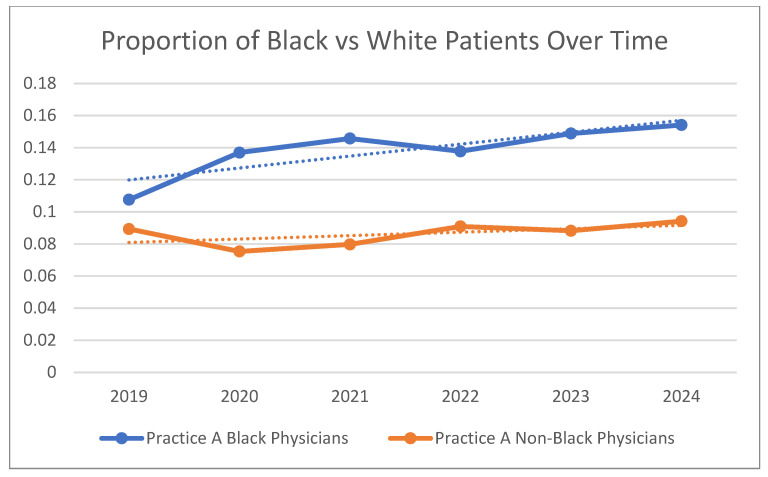
Percent of patient population self-reporting Black versus White race at first visit annually over time; comparing Black with other (nonBlack) physicians within Practice A. Dotted lines represent linear trend lines by Physician race.

**Table 1 healthcare-13-02255-t001:** Likelihood of patient self-reporting Black race at time of first visit annually from 2015–2024.

Year	Black Patients	Total Patients ^a^	Percent	Adjusted ^b^ Odds Ratio	95% Confidence Interval
Practice A					
2015	131	1825	7.2	Baseline	Baseline
2016	156	1939	8.1	1.15	0.89, 1.48
2017	179	2503	8.7	1.22	0.96, 1.56
2018	168	2077	8.1	1.15	0.90, 1.48
2019	233	2593	9.0	1.25	0.99, 1.58
2020	232	2705	8.6	1.17	0.92, 1.47
2021	318	3551	9.0	1.27	1.02, 1.59
2022	290	2927	9.9	1.46	1.17, 1.84
2023	275	2802	9.8	1.42	1.13, 1.79
2024	239	2327	10.3	1.49	1.17, 1.88
Practice B					
2015	156	2096	7.4	Baseline	Baseline
2016	154	2094	7.4	1.02	0.80, 1.30
2017	182	2243	8.1	1.15	0.91, 1.45
2018	174	2331	7.5	1.05	0.83, 1.33
2019	150	2589	5.8	0.84	0.66, 1.08
2020	139	2174	6.4	0.95	0.74, 1.22
2021	178	2599	6.9	1.04	0.82, 1.32
2022	130	2069	6.3	0.93	0.72, 1.20
2023	150	2149	7.0	0.97	0.76, 1.24
2024	83	1571	5.3	0.68	0.51, 0.91
Practice C					
2015	180	1902	9.5	Baseline	Baseline
2016	220	2011	10.9	1.22	0.98, 1.52
2017	193	1836	10.5	1.14	0.91, 1.43
2018	158	1611	9.8	1.11	0.88, 1.41
2019	125	1237	10.1	1.29	1.00, 1.67
2020	145	1387	10.5	1.21	0.95, 1.54
2021	107	1082	9.9	1.17	0.89, 1.52
2022	91	1088	8.4	0.95	0.72, 1.26
2023	158	1558	10.1	1.11	0.87, 1.41
2024 ^c^	82	872	9.4	1.16	0.87, 1.55

^a^ Limited population to only Black and White patients. ^b^ Adjusted for age, insurance type, area deprivation index, and count of visits outside of Practice A, B and C each year. ^c^ Through July 2024.

## Data Availability

Not available as data analyzed were accessed from the health system’s enterprise data warehouse.

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
