# Peer review of "Impact of Physician Practice Racial Composition on Patient Demographics"

_healthcare, 2025, doi:10.3390/healthcare13182255_

Round 1

Reviewer 1 Report (Previous Reviewer 1)

Comments and Suggestions for Authors

The abstract should state that Practices B and C had flat or decreasing trends.

In the introduction the secondary aim is about socioeconomic status (SES), but the measures are ADI and insurance. Briefly justify why these are good proxies for SES in this context.

The method of using a public website is acceptable but should be briefly defended or supplemented. Was this information verified with the physicians themselves or through HR records? A sentence acknowledging this potential limitation would strengthen this section.

For the results section, as a general rule of reporting the p values: if p value is greater than 0.05 should be reported with two decimal values, if p value is between 0.001 and 0.05 should be reported with three decimal places and if values shown on output as 0.000 should be reported as <0.0001; I suggest to make this correction in this section and throughout the paper.

In general, the central finding—that adding Black physicians increases the proportion of Black patients for the entire practice—is well-supported by the data

Author Response

Comment 1:The abstract should state that Practices B and C had flat or decreasing trends.

Response 1:Thank you for your comment. Abstract edited to reflect the trends.

Comment 2: In the introduction the secondary aim is about socioeconomic status (SES), but the measures are ADI and insurance. Briefly justify why these are good proxies for SES in this context.

Response 2: We have added to introduction that we used a proxy and further explain our rationale in methods: “While ADI doesn’t reflect individual patient circumstances, its measure of neighborhood level advantage impacts patients’ health outcomes”

Comment 3: The method of using a public website is acceptable but should be briefly defended or supplemented. Was this information verified with the physicians themselves or through HR records? A sentence acknowledging this potential limitation would strengthen this section.

Response 3: Thank you for this observation. We have added a sentence to limitation: “Another limitation is that while Black physicians’ race was known based on their inclusion on university website of minority physicians and their own self-report, self-identified race was not obtained for nonBlack physicians.”

Comment 4: For the results section, as a general rule of reporting the p values: if p value is greater than 0.05 should be reported with two decimal values, if p value is between 0.001 and 0.05 should be reported with three decimal places and if values shown on output as 0.000 should be reported as <0.0001; I suggest to make this correction in this section and throughout the paper.

Response 4: Thank you for this comment, we have made corrections throughout the text

Comment 5: In general, the central finding—that adding Black physicians increases the proportion of Black patients for the entire practice—is well-supported by the data

Response 5: Thank you

Reviewer 2 Report (Previous Reviewer 4)

Comments and Suggestions for Authors
  • Abstract reports p=0.1045; Results section reports p=0.233345 (independent calculation ≈0.221, suggesting abstract error or miscalculation).
  • Minor rounding discrepancies in slopes/p-values (e.g., Practice B slope -0.002 vs. calculated -0.0019; p=0.02693 vs. 0.029), but not major.
  • No adjustment for multiple comparisons across numerous p-values/tests, risking Type I errors.
  • Small effect sizes (e.g., slopes ~0.002-0.007) despite significance, due to large sample; clinical relevance modest.
  • No errors in verifiable linear regressions (slopes/p-values match closely via independent calc).
  • Utilization p-values (e.g., 0.002327 for Black in A) unverified without raw data, but consistent with trends.

Author Response

Comment 1: Abstract reports p=0.1045; Results section reports p=0.233345 (independent calculation ≈0.221, suggesting abstract error or miscalculation).

Response 1: Thank you for catching this. We have updated to p=0.23.

Comment 2: Minor rounding discrepancies in slopes/p-values (e.g., Practice B slope -0.002 vs. calculated -0.0019; p=0.02693 vs. 0.029), but not major.

Response 2: Thank you for catching this. We have updated to have consistent rounding throughout.

Comment 3: No adjustment for multiple comparisons across numerous p-values/tests, risking Type I errors.

Response 3: We chose not to adjust the p-value to avoid increasing the risk of Type II errors. We are open to using a bonferonni correction factor if the reviewers and editors prefer.

Comment 4: Small effect sizes (e.g., slopes ~0.002-0.007) despite significance, due to large sample; clinical relevance modest.

Response 4: We have added a sentence addressing this in our discussion: “Nonetheless, the fact that the practices do not provide the same type of clinical care was a limitation, and our large sample size contributes to statistical significance even for small effect sizes.”

Comment 5: No errors in verifiable linear regressions (slopes/p-values match closely via independent calc).

Response 5: In response to previous review we have added p-values in place of confidence intervals.

Comment 6: Utilization p-values (e.g., 0.002327 for Black in A) unverified without raw data, but consistent with trends.

Response 6: Thank you

Reviewer 3 Report (Previous Reviewer 3)

Comments and Suggestions for Authors

This study examines the impact of increasing the number of Black physicians on the racial composition of patients in gynecologic subspecialty clinics. The research employed a retrospective cohort design, selecting three clinics as study sites, with one clinic adding two Black physicians during the study period. The methodology is relatively sound, assessing the effect of physician racial composition on the patient population by comparing demographic characteristics—particularly race, insurance type, and socioeconomic status—across clinics. However, the study did not fully control for potential confounding factors, such as local outreach efforts, policy changes, or other improvements in healthcare services, which could also influence patients’ choices of where to seek care.

Regarding measurement indicators, the study used the proportion of Black patients, insurance type, and socioeconomic status (measured via the ADI index) as primary variables. These indicators are reasonably representative and operational, reflecting healthcare accessibility and equity. Nonetheless, the study did not deeply analyze the specific reasons patients chose particular physicians, such as whether racial concordance influenced their decision, nor did it measure patient satisfaction, adherence, or clinical outcomes—more direct indicators of healthcare quality—which somewhat limits the depth and applicability of the findings.

The results show that after adding Black physicians, the proportion of Black patients in Practice A increased significantly, and this effect was not limited to patients seen by the Black physicians. This finding is highly meaningful in practice, suggesting that physician racial diversity may generate a “spillover effect” that improves healthcare access for minority populations. Overall, the study is well-designed and data-rich, supporting the notion that increasing the number of Black physicians can enhance trust and engagement of Black patients in the healthcare system. Future research is recommended to expand to other specialties and adopt prospective designs to strengthen causal inference.

Author Response

Comment 1: This study examines the impact of increasing the number of Black physicians on the racial composition of patients in gynecologic subspecialty clinics. The research employed a retrospective cohort design, selecting three clinics as study sites, with one clinic adding two Black physicians during the study period. The methodology is relatively sound, assessing the effect of physician racial composition on the patient population by comparing demographic characteristics—particularly race, insurance type, and socioeconomic status—across clinics. However, the study did not fully control for potential confounding factors, such as local outreach efforts, policy changes, or other improvements in healthcare services, which could also influence patients’ choices of where to seek care.

Response 1: We mention briefly in our discussion that outreach from Black physicians may have impacted the results. “Any community outreach performed by Black physicians may also potentially lead to attracting more Black patients to the practice.”

 Even if this is the case, we believe it strengthens the impact of having Black physicians in a practice as they may be willing to do more outreach to patients.

Comment 2: Regarding measurement indicators, the study used the proportion of Black patients, insurance type, and socioeconomic status (measured via the ADI index) as primary variables. These indicators are reasonably representative and operational, reflecting healthcare accessibility and equity. Nonetheless, the study did not deeply analyze the specific reasons patients chose particular physicians, such as whether racial concordance influenced their decision, nor did it measure patient satisfaction, adherence, or clinical outcomes—more direct indicators of healthcare quality—which somewhat limits the depth and applicability of the findings.

Response 2: We agree and plan on exploring in future studies patient reasons for choosing their physicians.

Comment 3: The results show that after adding Black physicians, the proportion of Black patients in Practice A increased significantly, and this effect was not limited to patients seen by the Black physicians. This finding is highly meaningful in practice, suggesting that physician racial diversity may generate a “spillover effect” that improves healthcare access for minority populations. Overall, the study is well-designed and data-rich, supporting the notion that increasing the number of Black physicians can enhance trust and engagement of Black patients in the healthcare system. Future research is recommended to expand to other specialties and adopt prospective designs to strengthen causal inference.

Response 3: Thank you

This manuscript is a resubmission of an earlier submission. The following is a list of the peer review reports and author responses from that submission.

Round 1

Reviewer 1 Report

Comments and Suggestions for Authors

This study makes a contribution to understanding how physician diversity impacts patient demographics.

The abstract is structured; include specific numerical results (e.g., percentage increase in Black patients) for greater impact; the keywords should be checked in accordance with MeSH.

The introduction provides a strong rationale for the study, linking physician-patient racial concordance to improved health outcomes. Please refine the hypothesis to specify the expected mechanism (e.g., patient referrals, trust-building) / or provide reach questions (RQs).

In the methods section please address potential confounding factors (e.g., differences in practice specialties, patient recruitment strategies). How was patient race  self-reported and validated?

In the results section please report p-values or effect sizes for key comparisons (e.g., Black vs. non-Black physicians).

In the discussions section please provide more examples of specific frameworks that deal with patient matters in healthcare facilities, by referring to the scientific literature (fro e.g. doi: 10.3390/healthcare12030325 )

Please provide the conclusions of the study in a separate section, emphasizing  actionable steps for healthcare systems (e.g., diversity metrics for hiring).

The refines are few given the type of article and should be extended as suggested above to reach a minimum of 30. Please note the reference number with “[ ]” rather than “( )”.

Editing recommendation: the sections and subsections should be numbered.

Reviewer 2 Report

Comments and Suggestions for Authors

Brief summary

Please note I am not an expertise in epidemiology or statistics. I am a qualitative researcher.

Title: Impact of physician practice racial composition on patient demographics.

The central focus of this paper is to explore two key questions: (1) Does having a Black physical on staff increase the number of Black patients attending the practise? And (2) Does the addition of Black gynaecologists lead to increased visits from socioeconomical disadvantaged patients?”

Merit: This article contributes to the growing body of literature suggesting that the presence of Black physicians on staff positively influences the attendance and engagement of Black patients.

Accept after Minor Revisions: The paper can in principle be accepted after revision based on the reviewer’s comments.

General concept comments

The study received ethical approved by the University of Pittsburgh’s Institutional Review Board, ensuring appropriate oversight.

Patients data extracted using well-established markers for neighbourhood-level advantage, which supports the study’s methodological rigour.

Inclusion criteria are clearly defined, and race was recorded based on self-identification, aligning with best practices in demographic data collection.

The authors thoughtfully considered how to manage identifiable patient demographics, demonstrating ethical sensitivity in their approach.

The two comparative study groups addressed different gynaecological conditions within the boarder field of gynaecology, which is important context for interpreting the findings.

Specific comments 

General Note on the Introduction. Overall, the introduction provides a useful overview, but could benefit from more precise contextualisation for international readers.  

Line 32 – Consider adding greater specificity, especially given the journal’s international readership. It would be helpful to clarify that the evidence cited refers to a study conducted in the United States, specifically within the field of obstetrics and gynaecology.

Line 38 – 41 This section would be strengthened by including more specific details to support accessibility and relevance for a global audience.

Line 47 – The mention of the indirect cost of health care disparities is important. Expanding on this point would enhance the paper’s impact, particularly in relation to the economic implications of the American health system.

Line 33 – extra space, between contributor to … those disparities. Remove.

Line 47 – extra space, between health care …disparities. Remove.

Line 53 – extra space, practice… We. Remove.

Line 55 – extra space, practice… Because. Remove.

Line 174 – extra space, practice.(12)…This. Remove

Line 200 – extra space, practice.(3)…In. Remove

Line 220 – extra space, tokenism.(21,22)…Communities. Remove

Line 225 – extra space. quality-…of-life practices. Remove

While the paper offers valuable insights overall, certain sections – such as lines 226 to 232 – could benefit from deeper exploration. The brief treatment of these topics may reflect the authors’ decision to prioritize other focal areas, but further elaboration could strengthen the discussion.

Reviewer 3 Report

Comments and Suggestions for Authors

This study explores the impact of increasing the number of Black doctors on the proportion of Black patients in gynecological clinics, with significant social and public health implications. The study design is rigorous, employing a retrospective cohort research method to compare changes in the racial composition of patients in two gynecological clinics. One of the clinics added two Black doctors during the study period. The results show that the addition of Black doctors not only increased the proportion of Black patients in their personal consultations but also led to an overall rise in the proportion of Black patients in the entire clinic. This suggests that diversity among doctors may play a broader role in promoting healthcare equity.

Methodologically, the study controlled for several confounding factors such as patient age, insurance type, and area deprivation index, which enhanced the internal validity and interpretability of the findings.

However, the study still has certain limitations. First, the sample is limited to two gynecological sub-specialty clinics, which restricts the external generalizability of the results and makes it difficult to apply the findings to other specialties or primary care settings. Second, while the study controlled for various socio-economic variables, there may still be unmeasured confounding factors, such as patients' preferences for the race of their doctor or changes in the clinic's promotional strategies. Additionally, the study did not delve deeply into the reasons for the increase in the proportion of Black patients seen by non-Black doctors, leaving unclear whether this was due to patient choice or changes in the clinic's internal processes. These mechanisms warrant further exploration.

Overall, the study has strong practical value and provides solid empirical support for improving the accessibility of healthcare services for minority patients by increasing the proportion of minority doctors in the healthcare system. Future research could extend to a broader range of healthcare institutions and specialties, further exploring the mechanisms by which doctor diversity affects patient composition, in order to provide theoretical and practical guidance for formulating more effective healthcare equity policies.

Reviewer 4 Report

Comments and Suggestions for Authors
  1. The document contains significant repetition, particularly in the Methods section, where the same paragraph about comparing two gynecology practices is repeated multiple times. This suggests a formatting or editing error that obscures the full content and makes it difficult to assess the complete methodology.
  2. The Results section references multiple figures (Figure 1 to Figure 24) without providing their content or data, making it impossible to evaluate the trends described (“markedly increased” proportion of Black patients in Practice A). The absence of numerical data or statistical outputs limits the ability to verify the findings.
  3. The study does not clearly describe how physician race and sex were collected, how patient demographics were measured, or how socioeconomic disadvantage was quantified beyond the mention of the Area Deprivation Index (ADI). The imputation of missing ADI data (5% of visits) with the median ADI is mentioned but not justified statistically.
  4. The manuscript references “regressions” showing an increased likelihood of seeing Black patients in Practice A but does not specify the type of regression model used, the covariates included, or the statistical significance of the results. 
  5. The document contains inconsistencies in dates (“Healthcare 2021” vs. “Healthcare 2023” in headers) and typographical errors ( “FOR FEE REVIEW” and “FOR PEER REVIEW”). These errors suggest a lack of thorough proofreading.
  6. The document is truncated, particularly in the Results and later sections, limiting the ability to assess the full scope of the findings, discussion, and conclusions.